# Cryo-EM structures of MERS-CoV and SARS-CoV spike glycoproteins reveal the dynamic receptor binding domains

Yuan Yuan[1,2,*], Duanfang Cao[3,*], Yanfang Zhang[2,4,*], Jun Ma[3,*], Jianxun Qi[2], Qihui Wang[5], Guangwen Lu[6], Ying Wu[7], Jinghua Yan[5,8,9,10], Yi Shi[2,8,9,10], Xinzheng Zhang[3,11] & George F. Gao[1,2,4,8,9,10,12]

The envelope spike (S) proteins of MERS-CoV and SARS-CoV determine the virus host tropism and entry into host cells, and constitute a promising target for the development of prophylactics and therapeutics. Here, we present high-resolution structures of the trimeric MERS-CoV and SARS-CoV S proteins in its pre-fusion conformation by single particle cryo-electron microscopy. The overall structures resemble that from other coronaviruses including HKU1, MHV and NL63 reported recently, with the exception of the receptor binding domain (RBD). We captured two states of the RBD with receptor binding region either buried (lying state) or exposed (standing state), demonstrating an inherently flexible RBD readily recognized by the receptor. Further sequence conservation analysis of six human-infecting coronaviruses revealed that the fusion peptide, HR1 region and the central helix are potential targets for eliciting broadly neutralizing antibodies.

[1] School of Life Sciences, University of Science and Technology of China, Hefei, Anhui 230026, China. [2] CAS Key Laboratory of Pathogenic Microbiology and Immunology, Institute of Microbiology, Chinese Academy of Sciences, Beijing 100101, China. [3] National Laboratory of Biomacromolecules, CAS Center for Excellence in Biomacromolecules, Institute of Biophysics, Chinese Academy of Sciences, Beijing 100101, China. [4] Laboratory of Protein Engineering and Vaccines, Tianjin Institute of Biotechnology, Tianjin 300308, China. [5] CAS Key Laboratory of Microbial Physiological and Metabolic Engineering, Institute of Microbiology, Chinese Academy of Sciences, Beijing 100101, China. [6] West China Hospital Emergency Department (WCHED), State Key Laboratory of Biotherapy, West China Hospital, Sichuan University, and Collaborative Innovation Center of Biotherapy, Chengdu, Sichuan 610041, China. [7] School of Basic Medical Sciences, Wuhan University, Wuhan 430071, China. [8] Shenzhen Key Laboratory of Pathogen and Immunity, Shenzhen Third People's Hospital, Shenzhen 518112, China. [9] Center for Influenza Research and Early-warning, Chinese Academy of Sciences (CASCIRE), Beijing 100101, China. [10] Medical School, University of Chinese Academy of Sciences, Beijing 101408, China. [11] Center for Biological Imaging, CAS Center for Excellence in Biomacromolecules, Institute of Biophysics, Chinese Academy of Sciences, Beijing 100101, China. [12] National Institute for Viral Disease Control and Prevention, Chinese Center for Disease Control and Prevention (China CDC), Beijing 102206, China. * These authors contributed equally to this work. Correspondence and requests for materials should be addressed to Y.S. (email: shiyi@im.ac.cn) or to X.Z. (email: xzzhang@ipb.ac.cn) or to G.F.G. (email: gaof@im.ac.cn).

The emergence and persistence of Middle East respiratory syndrome coronavirus (MERS-CoV), almost one decade after the outbreak of severe acute respiratory syndrome coronavirus (SARS-CoV) in 2003, highlights the need for the rapid development of effective interventions against these highly pathogenic coronaviruses (CoVs). In 2002–2003, SARS-CoV first emerged in China and quickly spread to other countries, resulting in over 8,000 infected with ∼800 deaths[1]. MERS-CoV was first identified in the Middle East in 2012, specifically Saudi Arabia and Jordan[2,3]. Since then, MERS-CoV has reemerged on numerous instances in the Arabian Peninsula, occasionally spreading to other countries worldwide due to imported cases from travel[4–7]. Of note, in May 2015 a traveller returning from the Middle East caused a nosocomial outbreak of MERS in South Korea, involving 16 hospitals and 186 infected cases[8]. One of these cases then travelled to China, and accounted for China's only imported case thus far[4]. As of 25th July 2016, a total of 1,791 confirmed cases of MERS-CoV infection have been reported, including at least 640 related deaths in 27 countries[9]. As MERS-CoV grows in global importance, the World Health Organization (WHO) has prioritized it as one of eight pathogens to use as a blueprint to control and prevent newly emerging infectious diseases[10]. Moreover, SARS-CoV are still a threat to public health, as SARS-like CoV was found to circulate in bats[11].

Both MERS-CoV and SARS-CoV are zoonotic pathogens and are believed to have been transmitted from a natural host, possibly bats, to humans through intermediate mammalian hosts[12,13]. The key determinant of host specificity is the envelope-located trimeric spike (S) glycoprotein, which can be further cleaved by host proteases into an N-terminal S1 subunit and a membrane-bound C-terminal S2 region[14]. The cleaved S protein remains non-convalently associated in the metastable pre-fusion conformation. After virus endocytosis by the host cell, a second cleavage is generated, which is mediated by endo-lysosomal proteases (S2′ cleavage site), allowing membrane fusion activation to occur. In the S1 subunit, the receptor binding

domain (RBD, also called the C terminal domain, CTD) is localized in the C-terminal region, spanning ∼200 amino acids, and structural studies have revealed that the RBD consists of two subdomains: the core and external subdomains[14–17]. In the S2 subunit, the heptad repeat (HR) regions are also well characterized[18–20], and as expected, the HR1 and HR2 of MERS-CoV fold into an intra-hairpin helical structure that can assemble trimerically into a six-helix bundle (a trimer of the HR1/HR2 heterodimer), demonstrating a classical type I membrane fusion process[21]. Peptide inhibitors have been designed targeting these HR regions and been proven to be effective *in vitro* and *in vivo*[18,19,22–24]. These studies have provided insight into the characteristics of MERS-CoV and SARS-CoV S components; however, the overall S protein structures of these two highly pathogenic CoVs remain to be investigated. This will further enhance our understanding of S protein function and subsequent design of broadly neutralizing antibodies and vaccine immunogens.

Here we present high-resolution structures of the trimeric MERS-CoV and SARS-CoV S proteins in its pre-fusion conformation by single particle cryo-electron microscopy (cryo-EM). We captured two states of the RBD that is facilitated to receptor binding and further analysis of S proteins revealed the potential targets for eliciting broadly neutralizing antibodies.

## Results

**Overall structure of MERS-CoV and SARS-CoV S trimers.** We produced the MERS-CoV and SARS-CoV S trimers by fusing a T4 fibritin trimerization motif and 6X Histag at the C-terminal end of the ectodomain construct. For the MERS-CoV S protein, we also mutated the S2 cleavage site to enhance the stability. The resulting uncleaved MERS-CoV S ectodomain forms a trimer that can bind to the dimeric CD26 receptor protein, and then precipitate easily (Supplementary Fig. 1). We then used thrombin enzyme to remove the C-terminal T4 fibritin trimerization motif

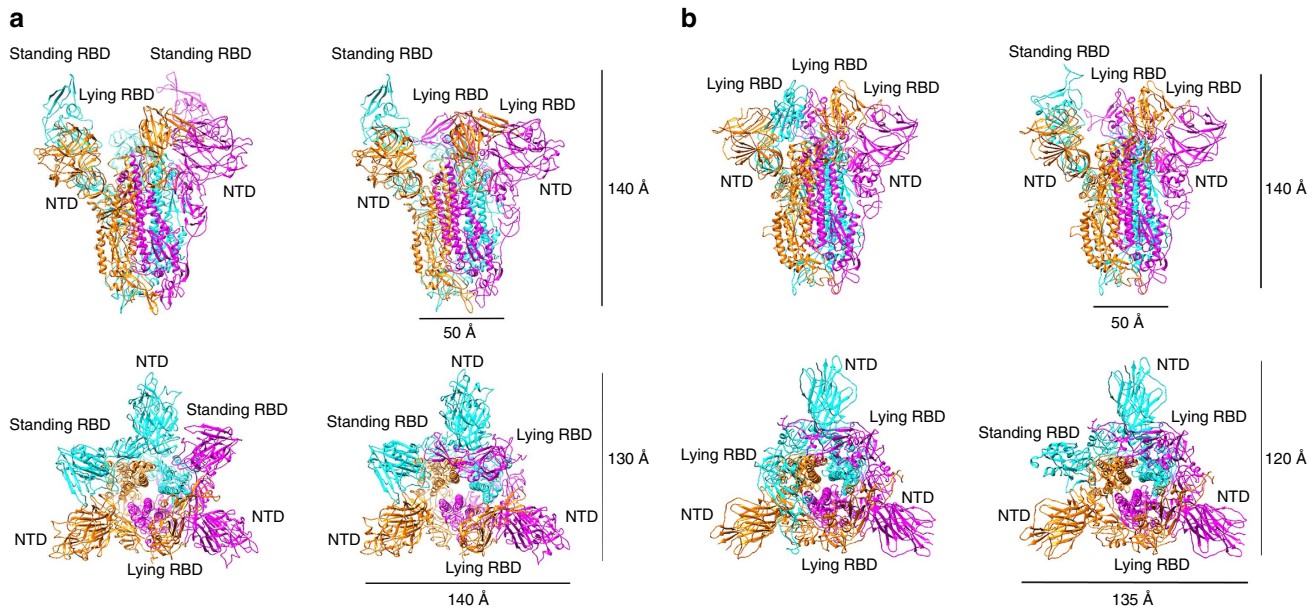

**Figure 1 | Overall structure of the MERS-CoV and SARS-CoV S ectodomain trimers.** (**a**) Two different conformations of the MERS-CoV S ectodomain trimer were determined without three-fold symmetry to resolutions of 4.1, 4.2 Å, respectively. The ribbon views of the structures are shown from both the side and the top. Two states (standing and lying) of the RBD were captured in the S ectodomain trimer structure. NTD domains are arranged in a triangular manner. (**b**) Two different conformations of SARS-CoV S ectodomain trimer were determined to resolutions of 3.2 and 3.7 Å, respectively. The ribbon views of the structures are shown from both the side and the top. Two states (standing and lying) of RBD were captured in the S ectodomain trimer structure. NTD domains are arranged in a triangular manner.

and 6X Histag, and found that the MERS-CoV S ectodomain trimer protein can be separated into two peaks in the gel filtration profile (Supplementary Fig. 2). One peak is the tag-removed MERS-CoV S ectodomain trimer, and the other is the mixed disassociated S1 and S2 subunits (Supplementary Fig. 2). The cleaved form of the MERS-CoV S ectodomain was confirmed by SDS–polyacrylamide gel electrophoresis (SDS–PAGE), and further N-terminal sequencing revealed that the S ectodomain protein was cleaved after residue R748 (Supplementary Fig. 3), three residues ahead of the S2 cleavage site. This indicated that once the MERS-CoV S ectodomain is cleaved into S1/S2 form, the S1 subunit tends to dissociate from S2. By contrast, the SARS-CoV S protein remains uncleaved after thrombin cleavage, and binds its receptor Angiotensin I Converting Enzyme 2(ACE2), confirmed by gel filtration survive assay (Supplementary Fig. 4).

Structures of both MERS-CoV and SARS-CoV S trimers were studied by single particle cryo-EM. Strikingly, for both trimers we observed two different classes of particles during three-dimensional classification, representing two conformations of the trimeric S protein with RBDs in different states (standing or lying) (Fig. 1).

For MERS-CoV S trimer, two classes were found with one or two of the three S1 RBDs in the S trimer in the 'standing' state. The reconstructed maps of these two conformations were refined to 4.1 and 4.2 Å resolutions without symmetry imposed, respectively (Supplementary Fig. 5). Aside from the RBD, the rest of the S1/S2 protein remained the same as in the trimer. To improve the resolution in the rest of the protein, we combined the data from both classes and determined the structure of MERS-CoV S ectodomain trimer with three-fold symmetry imposed at a resolution of 3.7 Å (Supplementary Fig. 5, see Methods). We also solved the crystal structure of the RBD-preceding N terminal domain (NTD, residues 18–353) in the S1 subunit at a resolution of 1.5 Å. An atomic model of the cleaved MERS-CoV S1/S2 trimer was built *de novo* using the 3.7 Å map, except for the flexible regions of S1 CTD and part of S1 NTD, which were fitted by crystal structures. The final model includes residues 18–1,206, with several small breaks due to the poor densities. The atomic model was used to

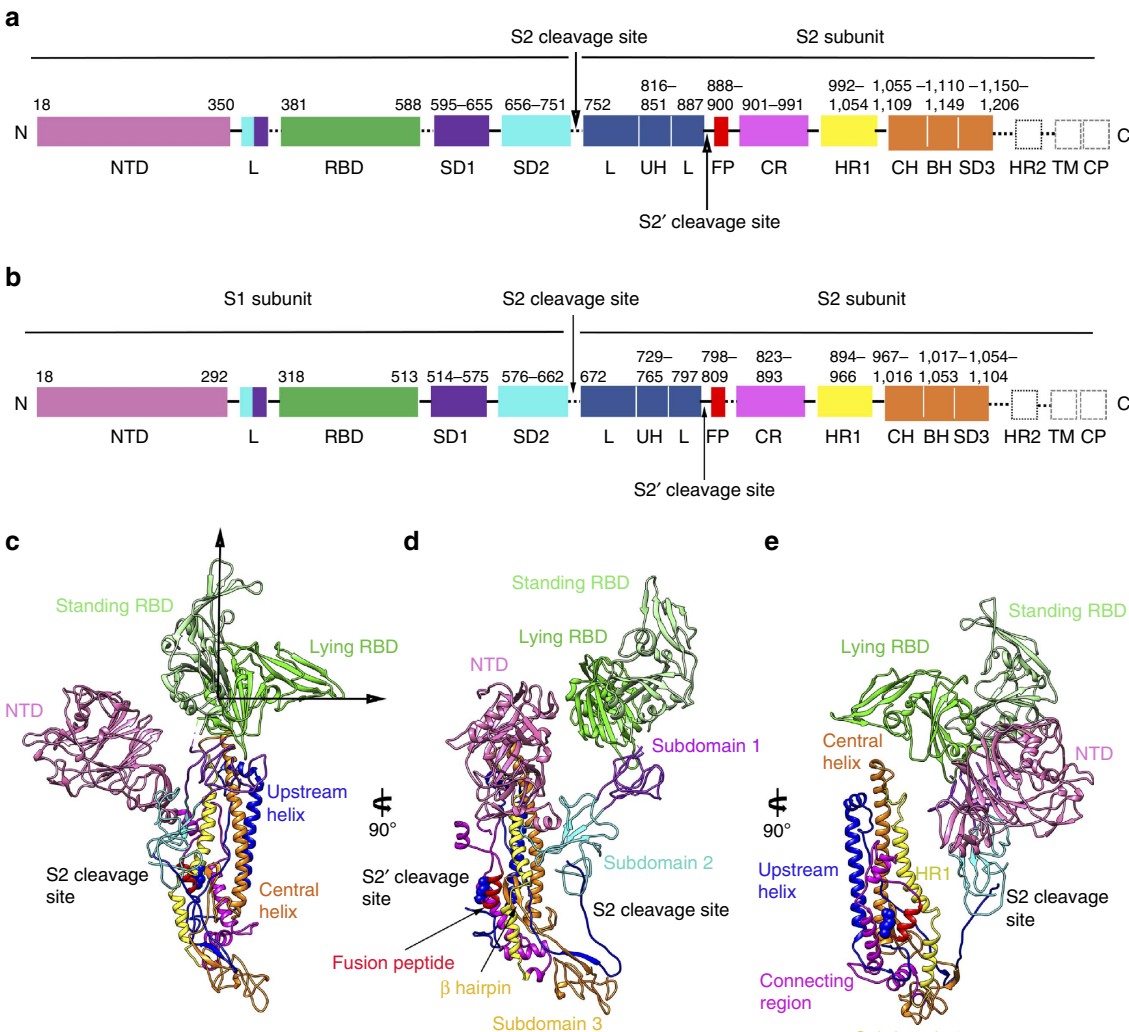

**Figure 2 | Architecture of the MERS-CoV and SARS-CoV S protomers. (a)** Schematic diagram of the MERS-CoV S glycoprotein organization. Black and grey dashed lines denote regions unresolved in the reconstruction and regions beyond the construct, respectively. NTD, N-terminal domain; L, linker region; RBD, receptor-binding domain; SD, subdomain; UH, upstream helix; FP, fusion peptide; CR, connecting region; HR, heptad repeat; CH, central helix; BH, β-hairpin; TM, transmembrane region/domain; CT, cytoplasmic tail. **(b)** Schematic diagram of the SARS-CoV S glycoprotein organization. The abbreviations of elements are the same as in **a**. **(c–e)** Ribbon diagrams depicting three views of the S protomer coloured as in **a**. As the MERS-CoV and SARS-CoV S protomers have extremely similar structures, and thus only MERS-CoV S protomer was used to show the detailed architecture.

interpret the 4.1 and 4.2 Å maps after being fitted into the map by domains.

For SARS-CoV S trimer, the structures of the two conformations with none or one of the three S1 RBDs in the 'standing' state were determined to resolutions of 3.2 Å (three-fold symmetry) and 3.7 Å (no symmetry), respectively (Supplementary Fig. 6). We also solved the crystal structure of the NTD at a resolution of 2.2 Å. An atomic model of the uncleaved SARS-CoV S trimer was built de novo using the 3.2 Å map, except the flexible RBD and part of the S1 NTD which were from fitted crystal structures. The final model includes residues 18–1,104, with several breaks due to the poor densities. The atomic model was used to interpret the 3.7 Å map after being fitted into the map by domains.

The MERS-CoV S ectodomain is a 140 Å long trimer with a triangular cross-section varying in diameter from 50 Å, at the membrane proximal base, to $140 \times 130$ Å at the membrane distal head (Fig. 1a), resembling a blooming flower. By contrast, the SARS-CoV S ectodomain has a smaller membrane distal head with dimensions of $135 \times 120$ Å (Fig. 1b). In the MERS-CoV or SARS-CoV S trimers, compared with the rest of the maps, the RBD region in the standing state features weaker and poorer density and has lower local resolution (Supplementary Figs 7 and 8), which likely correlates with the flexibility for receptor binding in vivo. By contrast, the NTD domain is observed with strong and clear density, forming a stable triangular platform on the top of the S trimer (Supplementary Figs 7 and 8). The flexible RBD regions are located on the triangular edges between the NTD domains (Supplementary Figs 7 and 8).

**Architecture dissection of MERS-CoV and SARS-CoV S trimers.** To date, little is known about the structural and functional information of NTDs for the MERS-CoV or SARS-CoV S proteins, though its counterparts from other CoVs, such as mouse hepatitis virus (MHV) and bovine coronavirus (BCoV), act as receptor binding domains and their crystal structures have already been delineated. Our crystal and cryo-EM structures show that MERS-CoV and SARS-CoV NTDs fold into galectin-like structures as in BCoV, MHV and HKU1CoV (Supplementary Figs 9 and 10). However, the glycan-binding site on the top of MERS-CoV NTD is occupied by a short helix and the N-linked glycan on that helix, and thus NTD in this conformation maybe unable to attach the cell surface by recognizing certain sugar molecules, unlike BCoV and HKU1 (refs 25,26). In addition to the NTD and RBD domains, the S1 subunits of both MERS-CoV and SARS-CoV contain two subdomains (I and II) that appear to be the base to underpin the NTD and RBD domains (Fig. 2). These two subdomains are primarily composed of amino acids following the RBD domain, and the linker region between the NTD and RBD, as well as residues adjacent to the S2 cleavage site, also contribute to the formation of the subdomains.

For both MERS-CoV and SARS-CoV S proteins, the S2 subunit is mainly composed of α-helices and forms the stem region of the S protein (Fig. 2). A long linker region connects the S2 cleavage site to the long upstream helix. The second S2′ cleavage site is exposed at the peripheral after the long upstream helix, and is readily accessible by the endo-lysosomal proteases (Fig. 2). An exposed helical fusion peptide is also observed immediately downstream of the S2′ cleavage site, and connects to the HR1 region by a long connecting region featuring three consecutive α-helices. Following the HR1 region, a long central helix stretches 70 Å along the three-fold axis towards the viral membrane (Fig. 2). This central helix is tightly packed against the upstream helix via hydrophobic contacts, forming the central stem region of the S ectodomain trimer with equivalent contributions from the other two S ectodomain protomers. After the central helix, a β-hairpin structure is present at the bottom of the S trimer. The

viral membrane proximal HR2 region is invisible due to poor density.

**Dynamic RBD domains of both S trimers.** Recently, three pioneering studies on cryo-EM structures of the S ectodomain from MHV and hCoVs HKU1 and NL63 have revealed a similar overall structural fold of the full-length S protein[27–29]. Both the MHV and HKU1 S ectodomain structures display a domain swapping organization of NTD and CTD in a woven appearance when viewed from the top of the S trimer. The CTD is located at the trimer apex close to the three-fold axis, whereas the NL63 S ectodomain structure shows a packed NTD and CTD organization (Supplementary Fig. 11). Unfortunately, these studies do not disclose how receptor binding occurs in SARS-CoV or MERS-CoV. The structural alignment of the SARS-CoV, MERS-CoV and NL63 CTD-receptor complexes with the S ectodomain structures reveals that the receptor binding surface of the CTD is buried in the S protein trimer (lying state) and is therefore incapable of making equivalent interactions without some initial breathing and transient conformational changes. However, our unprecedented observation of an inherently flexible RBD in both the MERS-CoV and SARS-CoV S trimers provides a plausible explanation for the receptor binding process for the two CoVs, as the receptor binding surface can be fully exposed in the standing state.

The MERS-CoV S1/S2 trimers could be classified into two classes with one (40% of particles) or two (60% of particles) RBDs in the standing state, and we cannot detect other conformations with all three RBDs in the standing or lying state. However, disassociated MERS-CoV S1 trimer particles were easily recognized during two-dimensional (2D) classification, which is consistent with the gel filtration result of the cleaved S protein as described above. We then reconstructed the cryo-EM structure of the disassociated MERS-CoV S1 trimer at a resolution of 9.5 Å (Fig. 3a, Supplementary Fig. 12). The disassociated S1 trimer forms a ring like structure, including the NTD domain, RBD domain and subdomains 1 and 2 (Fig. 3b). All three RBD domains are in a standing conformation (Fig. 3b). It implicates that the S1 trimer with three standing RBD domains is easily disassociated from the S2 moiety, and thus the stable S trimer particles with three standing RBD domains was rarely observed. Further analysis showed that the S1 trimer is stabilized by the interaction between the RBD core subdomain, subdomain 1 of one S1 protomer and the NTD domain of the neighbouring S1 protomer (Fig. 3c,d).

The SARS-CoV S trimer can be classified into two classes with all three RBDs in the lying state (56% of particles) or two lying RBDs and one standing RBD (44% of particles). Combined with MERS-CoV S1/S2 trimer, we have shown that the RBD is indeed flexible in highly pathogenic CoVs.

**Implication for the design of broadly neutralizing antibodies.** The S protein is the major antigen on the surface of the MERS-CoV or SARS-CoV virion. For MERS-CoV, most of the currently-available neutralizing antibodies were developed against the flexible RBD region[30–34]. For SARS-CoV, neutralizing antibodies against both S1 and S2 subunits have been developed[35,36]. The accessibility of the RBD domain, due to its inherent flexibility, as shown in this study provides an explanation for the high efficiency of RBD-directed neutralizing antibodies. Since the RBD is located between the NTD domains, a strategy to develop neutralizing antibodies, which target the NTD and interfering with receptor binding through steric hindrance should be feasible in the future.

Previous studies showed that N-linked glycosylation in the viral envelope protein can help the virus evade immune

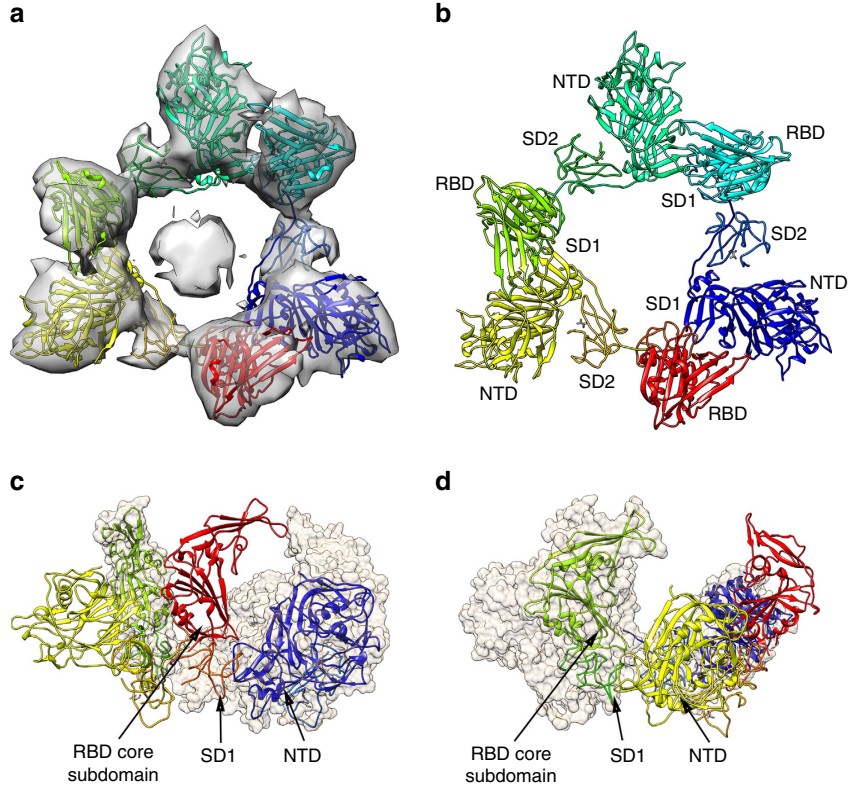

**Figure 3 | Low resolution cryo-EM structure of the disassociated S1 trimer.** (**a**) Cryo-EM electron density of the overall structure of the disassociated S1 trimer. (**b**) Ribbon view of the S1 trimer, including NTD, RBD, SD1 and SD2. (**c**,**d**) Interaction between S1 protomers. The RBD core subdomain and SD1 form quaternary interactions with the NTD to stabilize the disassociated S1 trimer.

surveillance. Therefore, we analysed the N-linked glycosylations of MERS-CoV and SARS-CoV S trimers. In the cryo-EM reconstruction, we observed the density for 10 N-linked glycans in MERS-CoV S protein and 14 N-linked glycans in SARS-CoV S protein (Fig. 4a,b). In fact, the MERS-CoV S protein has 25 potential N-linked glycosylation sites, and SARS-CoV possesses 22 potential N-linked glycosylation sites (Fig. 4c,d). Most of the N-linked glycosylation sites are located on the S1 subunit and the C-terminal region (including HR2 region and the region preceding HR2) of S2 subunit (Fig. 4c,d). For FR, HR1 region and central helix, there are no N-linked glycosylation sites (Fig. 4c,d). Further conservation analysis of full-length sequences of the S protein from six human-infecting CoVs (MERS-CoV, SARS-CoV, HKU1, NL63, OC43 and 229E) revealed that the glycosylation variable regions are mainly located on the S1 subunit, including the NTD and RBD regions, whereas the S2 subunits are relatively conserved (Fig. 4e,f). It is worth to note that the fusion peptide (FP) and HR1 region are exposed at the surface of stem region of the S trimer, and provide a patch of conserved region for epitope-focused vaccine immunogen design aimed at raising broadly neutralizing antibodies against human-infecting CoVs (Fig. 4e,f). In addition, the flexible RBD regions allow the top of S1 in an open state, and enable the central stem region of the S trimer, including the top region of the upstream helix, HR1 and central helix, to become accessible to antiviral protein inhibitors (Fig. 4e,f).

**Discussion**

Here we show that both MERS-CoV and SARS-CoV S trimers have flexible RBD, and then we further constructed the receptor binding models for the MERS-CoV or SARS-CoV S trimers by superimposition of the S trimer structures with the RBD-receptor complex structures through the RBD domain (Fig. 5). We hypothesis that on the cell surface one CD26 may crosslink two S trimers by binding to standing RBDs, one from each trimer, whereas the monomeric ACE2 receptor will bind to the SARS-CoV S trimer in the pattern of one receptor to one S trimer (Fig. 5). Thus, MERS-CoV might have higher avidity to receptor binding than SARS-CoV, when these two CoVs are attached to the host cell surface.

The spatial organizations of MERS-CoV and SARS-CoV S proteins resemble that of influenza virus hemagglutinin (HA) protein, which also has two cleaved subunits (HA1 and HA2) and the HA1 subunit must dissociate from HA2 before activation of membrane fusion under low pH environment in the endosome[21]. A feasible membrane fusion process of MERS-CoV and SARS-CoV is proposed bellow. Taking MERS-CoV as an example, the receptor binding to the RBD region may help to keep the RBD in the 'standing' state, which facilitates the dissociation of the S1 subunit from the S2 subunit. When the S1 subunit is dissociated from the S2 subunit (Fig. 6), a second S2′ cleavage can release the fusion peptide. The connecting region, HR1 region and central helix would form an extremely long helix (at least 200 Å) to insert the fusion peptide into the host cell membrane (Fig. 6), which is deduced from the fusion process of the influenza HA protein. Finally, the HR1 and HR2 regions will form a coiled structure and assemble into a six-helix bundle to drag the viral and host membranes together (Fig. 6).

In summary, the observation of flexible RBD in MERS-CoV and SARS-CoV S proteins has an important implication for the pathogenesis: for these two CoVs, the flexible RBD can readily be approached by the receptors to bind and guarantee virus entry. Our results have provided an important framework to understand

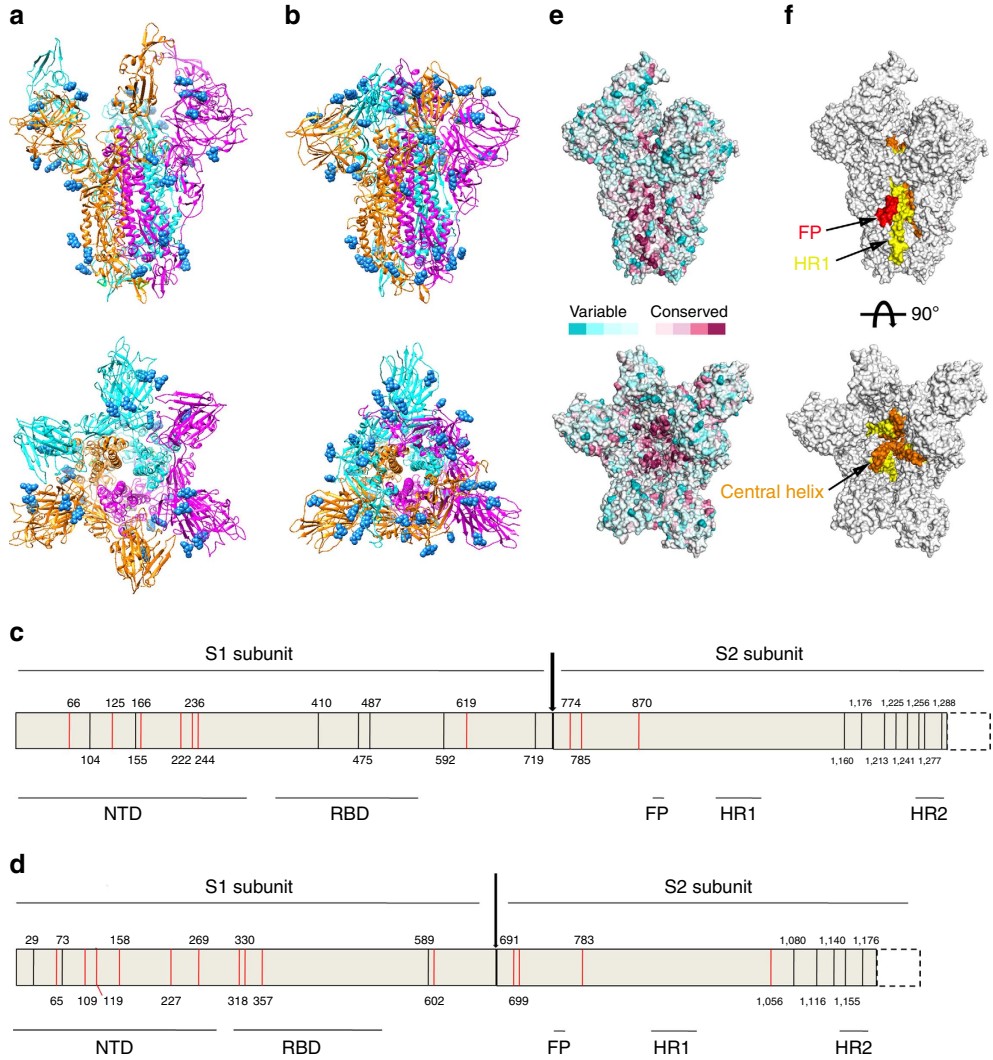

**Figure 4 | N-linked glycosylation analysis of MERS-CoV and SARS-CoV S proteins and a potential strategy for antiviral intervention.** (**a,b**) Cartoon representation of MERS-CoV (**a**) and SARS-CoV (**b**) S trimers from the side and top views. The glycans are shown in spheres. (**c,d**) Schematic diagram of the N-linked glycosylation sites for MERS-CoV (**c**) and SARS-CoV (**d**) S proteins. The visible N-linked glycosylation sites are shown in red lines and the invisible N-linked glycosylation sites are shown in black lines. (**e**) Surface representation of the MERS-CoV S trimer from either the side or the top, coloured according the sequence conservation from the most conserved (magenta) to the most divergent (cyan), using the ConSurf server based on an alignment of S sequences from six human-infecting CoV in the NCBI database. (**f**) Surface representation of the MERS-CoV S trimer highlighting the highly conserved region for the design of broadly neutralizing antibodies, including exposed FP, HR1 region and central helix.

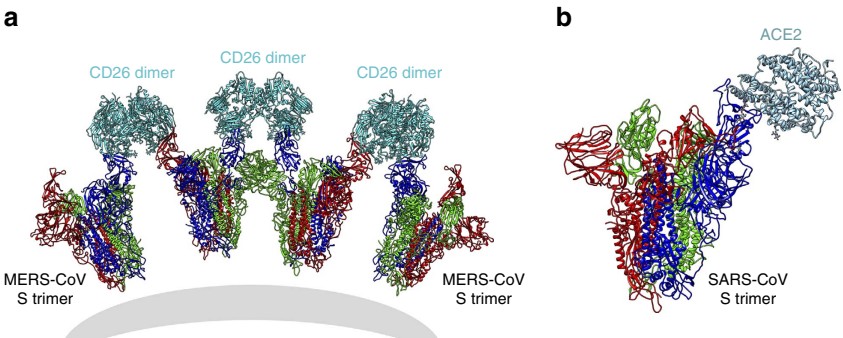

**Figure 5 | Models of MERS-CoV and SARS-CoV S trimers bound to their receptors.** The models were built by superimposition of the S trimer structures with the RBD-receptor complex structures through the RBD domains. The MERS-CoV S trimer can cross-link the dimeric CD26 receptor during the binding (**a**), whereas the SARS-CoV S trimer can bind one monomeric ACE2 receptor (**b**).

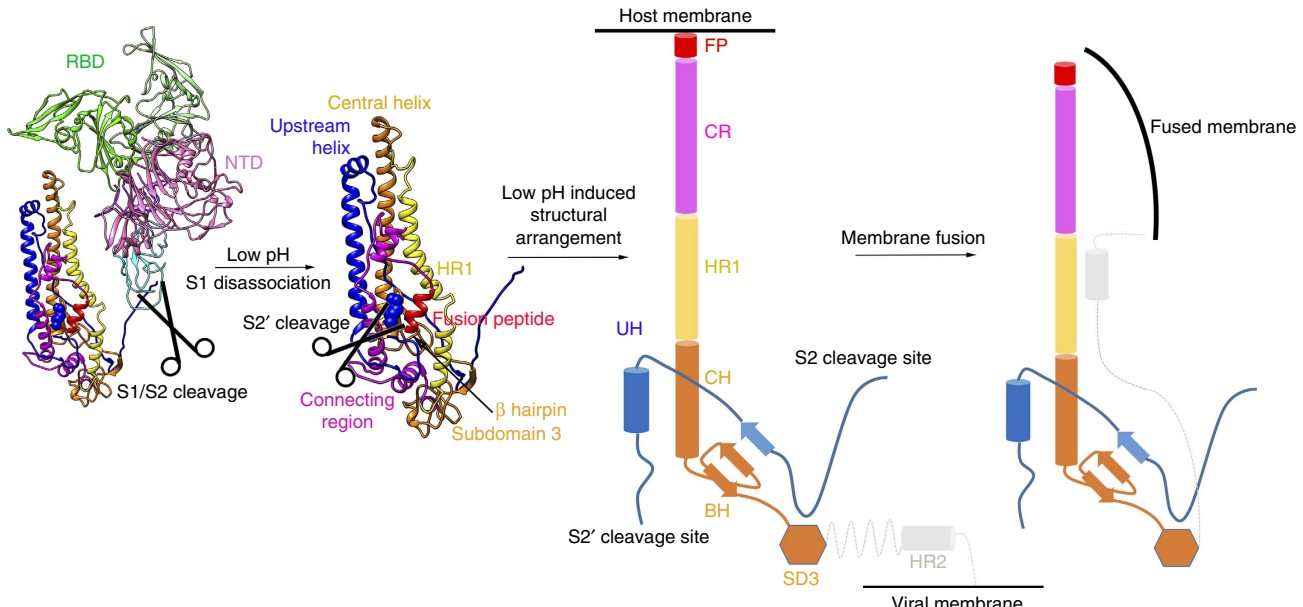

**Figure 6 | Proposed mechanism of membrane fusion promoted by MERS-CoV S protein.** After cleavage into S1/S2 subunits, the S1 subunit is easily disassociated from the S2 subunit. In the endosome, the S2′ cleavage site could be further cleaved by the host proteases, releasing its fusion peptide. Then, under low pH environment, the connecting region, HR1 helix and central helix undergo structural rearrangement to form a long helix to help the insertion of the fusion peptide into the host membrane. Finally, the HR1 and HR2 fold into an intra-hairpin helical structure that can trimerically assemble into a six-helix bundle, resulting in membrane fusion.

the entry mechanisms of MERS-CoV and SARS-CoV, and suggest ways for preventing or controlling future outbreaks of MERS-CoV and SARS-CoV.

## Methods

**Protein expression and purification.** The gene encoding MERS-CoV spike protein (GenBank accession number JX869059, residues 18–1,294, with an Arg751Ser mutation to abolish the protease cleavage site) and the SARS-CoV spike gene (GenBank accession number AY2,78,488, residues 14–1,193) were both synthesized and subcloned into the baculovirus transfer vector pFastbac1 (Invitrogen) with a N-terminal gp67 signal peptide, a C-terminal thrombin cleavage site followed by a T4 fibritin trimerization domain and a 6X Histag. The two kinds of S protein were produced with Bac-to-Bac baculovirus expression system (Invitrogen) separately. Transfection and virus amplification were conducted with Sf9 cells, and Hi5 cells (Invitrogen) were used to produce the recombinant proteins. Soluble S protein was captured from cell supernatants by metal affinity chromatography using a HisTrap HP 5 ml column (GE Healthcare). The eluted product was pooled and further purified by gel filtration chromatography with a Superose 6 10/300 GL (GE Healthcare) column equilibrated with a buffer containing 20 mM Tris–HCl (pH7.5) and 150 mM NaCl. Then, the S proteins were both cleaved with thrombin (Sigma, 3 units per mg S protein) at 4 °C overnight to remove the C-terminal trimerization domain and 6 × His-tag. A final round of size exclusion chromatography was conducted to purify the cleaved product with a Superose 6 10/300 GL column. The resulting S proteins reached a purity of 95% as shown by SDS–PAGE (Supplementary Figs 2 and 4).

The coding sequence for N terminal domain (NTD, spanning residues 18–353) of MERS-CoV S protein (MERS-CoV S-NTD) was cloned into the *Eco*RI and *Xho*I restriction sites of pFastBac1 vector for baculovirus expression (Bac-to-Bac baculovirus expression system, Invitrogen). An N-terminal gp67B signal peptide and a C-terminal 6X Histag were added to facilitate protein secretion and purification. The MERS-CoV S-NTD protein was purified by Ni-NTA affinity column and Superdex200 gel filtration column (GE Healthcare). The protein was concentrated to 15 mg ml$^{-1}$ in buffer containing 20 mM Tris, pH 8.0 and 150 mM NaCl for crystal screening. The SARS-CoV S-NTD (spanning residues 14–292) was constructed, expressed and purified with the same strategy.

**N-terminal sequencing.** The thrombin-cleaved MERS S protein was separated by SDS–PAGE and subsequently electroblotted to polyvinylidene fluoride membrane with CAPS buffer (10 mM CAPS, pH 11, 10% methanol) at 200 mA for 1.5 h. The polyvinylidene fluoride membrane was stained with freshly prepared Coomassie Blue R250 (0.1% Coomassie Blue R250, 1% acetic acid, 40% methanol) for 50 s and destained with 50% methanol until bands were visible and the background was

clear. Then the membrane was dried and the target bands were cut for the N-terminal sequencing with the Edman degradation method by using PPSQ-31A (Shimadzu Corporation, Japan).

**Crystallization and structure determination.** The monomeric MERS S-NTD was crystallized by the sitting-drop vapour diffusion method at 18 °C with 1 µl protein solution mixed with 1 µl reservoir buffer. High-quality crystals of MERS S-NTD grew in buffer of 0.2 M Magnesium chloride hexahydrate, 0.1 M BIS-TRIS pH 5.5, 25% w/v Polyethylene glycol 3,350at a protein concentration of 15 mg ml$^{-1}$. Derivative crystals were obtained by soaking MERS S-NTD crystals overnight in mother liquor containing 2 mM KAuCl4. The SARS S-NTD was also crystallized by the sitting-drop vapour diffusion method at 18 °C. High-quality crystals grew in 1.3 M Na/K hydrogen phosphate (pH 7.0) at a protein concentration of 15 mg ml$^{-1}$. Diffraction data were collected with cryoprotected (in a reservoir solution containing 20% [v/v] glycerol) crystals at the Shanghai Synchrotron Radiation Facility beamline BL17U. All the datasets were processed with HKL2,000 software[37]. The structure of MERS S-NTD was determined by the SAD method using Au derivative data set with SHELXD (ref. 38) and Phaser-ep (ref. 39), while the structure of SARS S-NTD was determined by the molecular replacement method using cryo-EM structure. The atomic model was completed with Coot[40] and refined with phenix.refine in Phenix[41], and the stereochemical quality of the final model was assessed with Molprobity[42]. Data collection, processing and refinement statistics are summarized in Supplementary Table 1. The native data set was collected at 0.979 Å, while the derivative data set was collected at 1.039 Å.

**Cryo-electron microscopy data collection and processing.** Purified S protein (3 µl) with a concentration of ∼0.4 mg ml$^{-1}$ for MERS-CoV S or ∼0.3 mg ml$^{-1}$ for SARS-CoV S was placed on a glow-discharged holy carbon grid (GIG, 1.0 µm hole size, 400 mesh). After 4 s blotting with filter paper, the grid was flash plunged in liquid ethane using an automatic plunge device (Leica EM GP) with 10 °C temperature and 99% humidity. Cryo-EM single particle data collection was performed using a 300 kV Titan Krios microscope equipped with K2 camera. Using the super resolution mode, each image was exposed of 11 s at a calibrated magnification of 38461 and an electron dose rate of ∼8 e per pixel per s, resulting in a total dose of ∼50 e Å$^{-2}$ that was fractionated into 32 movie frames. The images were binned before data processing, yielding a final pixel size of 1.3 Å.

In each micrograph, after beam induced motion of each movie frame being corrected by the program MOTIONCORR (ref. 43), a 32-movie frames averaged micrograph was calculated and the parameters of the contrast transfer function on this micrograph was determined by the program ctffind[44]. A subset of protein particles were semi-automatically boxed using the program e2boxer.py in EMAN2 software package[45] and processed with 2D classification. Automatic particle boxing of the whole data set was performed by RELION program,[46] using previously

obtained three distinguished class average images as references. A total number of ~530,000 particles were picked in 1,810 micrographs and processed by no reference 2D classification using RELION program. About 260,000 particles in the good classes representing the S1/S2 trimers (Supplementary Fig. 4a) were kept for further 3D classification. The HKU1 S trimer density map was low-pass filtered to 60 Å and rescaled as a reference map for 3D classification without imposing any symmetry. All the particles were classified into six classes and a 3D model within each class was reconstructed. Among the six reconstructions, two of them having the most accurate rotational alignment have reasonable rod-like densities in the middle representing central helices of S2. Class one containing about 55,000 particles has two RBDs in a standing state and one RBD in a lying state. Class two containing about 40,000 particles has one RBD in a standing state and two RBDs in a lying state. Further classification could not identify other conformations such as all three RBDs in a standing state or all three RBDs in a lying state, probably because of the small population of particles with these conformations. The rest part of the S protein monomer kept the same in a trimer. Thus, for better alignment, a 4.1 Å reconstruction with three-fold symmetry imposed containing all the particles from these two classes was calculated. However, the density of RBD region became quite low due to the average of RBD density between lying state and standing state. Further particle based motion correction and particle shinning process improve the resolution to 3.7 Å of the three-fold symmetry reconstruction by 0.143 criterion in the gold standard Fourier Shell correlated Coefficient (Supplementary Fig. 4d). In addition, the shiny particles in class one and two were used to calculate a 4.1 and a 4.2 Å map without imposing any symmetry, respectively. The orientation distribution (Supplementary Fig. 4c) of MERS S protein trimer in the three-fold symmetry reconstruction was similar to that of HKU1 (ref. 27). The local resolution of the three maps was calculated using program ResMap[47].

The data of SARS-CoV S was processed in the same way as mentioned above, and the shiny particles in class one and two were used to calculate a 3.2 Å with three-fold symmetry imposed and a 3.7 Å map without imposing any symmetry.

During 2D classification of MERS-CoV S protein data, some of the class-averaged images had a hole in the middle of the protein. We selected the particles (~60,000) within these classes for the reconstruction of S1 trimers. These class-averaged images were used to build an initial model of S1 trimer for 3D classification by e2initialmodel.py program[45]. After 3D classification, ~15,500 particles were kept for the high-resolution refinement imposing the three-fold symmetry which resulted in a 9.5 Å map of S1 trimer. We were not able to identified separate S2 proteins probably because S2 trimer was lacking of a stable conformation.

**Model building and refinement.** For model building, the predicted model of MERS-CoV or SARS-CoV S protein from the Phyre2 web server[48] was used as the starting model. De novo building was performed manually in COOT (ref. 49) based on the well-defined continuous electron density of its main chain in the three-fold symmetry map, and sequence assignment was guided mainly by bulky amino-acid residues densities. For MERS-CoV S model, the NTD domain and RBD domain were generated by fitting its crystal structure into the electron density map. For SARS-CoV S model, the NTD initial model was built manually in COOT based on the electron density of its main chain in the map, and then the NTD initial model was used as template for crystal structure determination. Finally, the model of SARS-CoV S NTD domain and RBD domain were generated by fitting the crystal structures into the electron density map. The structure model was first refined in real space against the cryo-EM map using phenix.real_space_refine application in PHENIX (ref. 50) with geometry and secondary structure restraints. Refinement in reciprocal space was then performed in REFMAC (ref. 51) with stereo-chemical. Automatic real-space and reciprocal-space refinements followed by manual correction in COOT were carried out iteratively until there were no more improvements in both R factor and geometry parameters. The refinement statistics of the structural model are summarized in Supplementary Table 2. For the reconstructions of MERS-CoV S class one, class two, S1 trimer and SARS-CoV S class one, class two, all domains of S1 and S2 model were fitted into the corresponding maps separately.

**Data availability.** Coordinates and structure factors of the crystal structures reported here have been deposited into the Protein Data Bank: MERS-NTD (PDB code: 5X4R), SARS-NTD (PDB code: 5X4S). Coordinates and cryo-EM maps of SARS-CoV and MERS-CoV S trimers have been deposited into the Protein Data Bank: SARS-CoV S conformation 1 (PDB codes: 5X58, EMD-6,703), SARS-CoV S conformation 2 (PDB codes: 5X5B, EMD-6,705); MERS-CoV S with three-fold symmetry (PDB codes: 5X59, EMD-6,704), MERS-CoV S conformation 1 (PDB codes: 5X5C, EMD-6,706), MERS-CoV S conformation 2 (PDB codes: 5X5F, EMD-6,707). All other relevant data are available from the corresponding authors on reasonable request.

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

## Acknowledgements

We thank X.J. Huang, G. Ji, W. Ding, F. Sun and other staff members at the Center for Biological Imaging (IBP, CAS) for their assistance in data collection. This work is supported by the Strategic Priority Research Program of the Chinese Academy of Sciences (Grant No. XDB08020100), National 973 Project (Grant No. 2013CB531502 and 2014CB542503) of Ministry of Science and Technology (MOST) of China, the National Key Research and Development Program of China (Grant No. 2016YFD0500300), the China National Grand S&T Special Project (No. 2014ZX10004-001-006) and the Natural Science Foundation of China (NSFC, Grant No. 31570874 and 81461168030). Y.S. is supported by the Excellent Young Scientist Program from the NSFC (Grant No. 81622031), the Excellent Young Scientist Program of the Chinese Academy of Sciences and the Youth Innovation Promotion Association CAS (2015078). X.Z. received scholarships from the 'National Thousand (Young) Talents Program' from the Office of Global Experts Recruitment in China. G.F.G. is a leading principal investigator of the NSFC Innovative Research Group (Grant No. 81621091).

## Author contributions

Y.S., X.Z. and G.F.G designed and supervised the project. Y.Y., D.C., Y.Z. and J.M. conducted the experiments. J.Q. collected the data set and solved the crystal structures. Y.S., X.Z. and G.F.G. analysed the data and wrote the manuscript. Q.W., G.L., J.Y. and Y.W. participated in the discussion and manuscript editing.

## Additional information

**Competing interests:** The authors declare no competing financial interests.

