## [Peer Review File · Nature Communications]

PEER REVIEW FILE

Reviewers' Comments:

Reviewer #1 (Remarks to the Author):

This manuscript describes cryo-EM structures of SARS and MERS S trimers. The main finding is that the receptor binding domain (RBD) is flexible and can adopt two different conformations, lying and standing. In the standing conformation, the domain can conceivably bind the receptor. This is a novel and important observation and true for both characterised coronaviruses. The structures allow the authors to suggest strategies for immunogen / therapeutic antibody design development.

The manuscript appears now technically sound. Further classification of the cryo-EM data in different structural states has improved the results, proper statistics have been added for resolution estimates, and angular distributions are shown.

In its current form the manuscript is difficult to read; many sentences are too long and confusing. Parts appear somewhat hastily written, especially where results on SARS-CoV have been added. Text should be carefully edited for clarity and grammar. Results should be split in shorter paragraphs and each section should have a descriptive subtitle (subtitles like "fusion machinery" do not help the reader much).

Deductions from the structures should be made with more care. The statement that "We found that the dimeric CD26 receptor cross-links the MERS-CoV S trimer with two standing RBDs" is too strongly worded as there is no experimental evidence for this. Better to say "we hypothesise that on the cell surface one CD26 may cross-link two S trimers by binding to two standing RBDs, one from each trimer"

Another example of such a too strong deduction is "the glycan-binding site on the top of MERS-CoV NTD is occupied by a short helix and the N-linked glycan on that helix, preventing initial viral attachment to the cell surface by recognising certain sugar molecules". Better would be to say "S protein in this conformation may be unable to attach"

The description of S structures from MERS and SARS are repetitive in the text and so are legends to Fig. 2 and 3. The authors should avoid such repetition and consider combining these.

After the manuscript has been formatted to the style of the journal (separate Results and Discussion), it would be best to leave speculations related to Fig. 5 and 6 for Discussion (Fig 7. e.g. analysis of N-linked glycans clearly belongs to Results so it should come before these).

Reviewer #2 (Remarks to the Author):

The authors have addressed all of my concerns in the revised manuscript and I feel that it is improved and appropriate for publication.

Point-by-point response letter

Referee #1 (Remarks to the Author):

1. This manuscript describes cryo-EM structures of SARS and MERS S trimers. The main finding is that the receptor binding domain (RBD) is flexible and can adopt two different conformations, lying and standing. In the standing conformation, the domain can conceivably bind the receptor. This is a novel and important observation and true for both characterised coronaviruses. The structures allow the authors to suggest strategies for immunogen / therapeutic antibody design development.

The manuscript appears now technically sound. Further classification of the cryo-EM data in different structural states has improved the results, proper statistics have been added for resolution estimates, and angular distributions are shown.

In its current form the manuscript is difficult to read; many sentences are too long and confusing. Parts appear somewhat hastily written, especially where results on SARS-CoV have been added. Text should be carefully edited for clarity and grammar. Results should be split in shorter paragraphs and each section should have a descriptive subtitle (subtitles like "fusion machinery" do not help the reader much).

RESPONSE:

Thank you for your kind suggestion. The manuscript has been extensively edited for clarity and grammar by a native English speaker. Results has been split in shorter sections as you suggest and subtitles have been added to each sections.

2. Deductions from the structures should be made with more care. The statement that "We found that the dimeric CD26 receptor cross-links the MERS-CoV S trimer with two standing RBDs" is too strongly worded as there is no experimental evidence for this. Better to say "we hypothesise that on the cell surface one CD26 may cross-link two S trimers by binding to two standing RBDs, one from each trimer"

RESPONSE:

Thank you for your suggestion. We have modified the description in the revised manuscript.

3. Another example of such a too strong deduction is "the glycan-binding site on the top of MERS-CoV NTD is occupied by a short helix and the N-linked glycan on that helix, preventing initial viral attachment to the cell surface by recognising certain sugar molecules". Better would be to say "S protein in this conformation may be unable to attach"

RESPONSE:

Thank you for your suggestion. We have modified the description in the revised manuscript.

4. The description of S structures from MERS and SARS are repetitive in the text and so are legends to Fig. 2 and 3. The authors should avoid such repetition and consider combining these.

RESPONSE:

Thanks for your kind suggestion. We have revised the description of S structures from MERS and SARS in the text and combined Fig.2 and 3 together.

5. After the manuscript has been formatted to the style of the journal (separate Results and Discussion), it would be best to leave speculations related to Fig. 5 and 6 for Discussion (Fig 7. e.g. analysis of N-linked glycans clearly belongs to Results so it should come before these).

RESPONSE:

Thanks for your advice. In the revised manuscript, we have left the speculations related to Fig.5 and 6 for discussion and the analysis of N-linked glycans related to Fig.7 come before them.

Reviewer #2 (Remarks to the Author):

The authors have addressed all of my concerns in the revised manuscript and I feel that it is improved and appropriate for publication.

RESPONSE:

Thanks for your positive feedback. We appreciate your comment and help to improve the paper.